# Welfare Implications for Hares, *Lepus timidus hibernicus*, Taken from the Wild for Licensed Hare Coursing in Ireland

**DOI:** 10.3390/ani10010163

**Published:** 2020-01-17

**Authors:** Andrew Kelly

**Affiliations:** 1Irish Society for the Prevention of Cruelty to Animals, National Animal Centre, Keenagh, Co. Longford N39 X257, Ireland; andrew.kelly@ispca.ie; 2School of Biological Sciences, Queen’s University Belfast, Belfast BT7 1NN, UK

**Keywords:** hare, Irish hare, *Lepus timidus hibernicus*, coursing, greyhound, animal welfare

## Abstract

**Simple Summary:**

Hare coursing is a widespread and controversial activity in the Republic of Ireland in which the speed and agility of greyhounds are measured against that of a hare. Every year, several thousands of hares are taken from the wild by coursing clubs under licenses issued by the National Parks and Wildlife Service. While the majority are returned to the wild at the end of the coursing season, no research has been done on the impact of hare coursing on the welfare of individual hares. Despite greyhounds in licensed coursing events being muzzled since 1993, hares may be pinned to the ground by the dogs and killed or so severely injured that they have to be euthanized by a veterinary practitioner. In addition to animal welfare concerns, the recent emergence in Ireland of rabbit haemorrhagic disease (RHD2) has led to calls for licensed hare coursing to be prohibited on animal welfare, disease control, conservation and ethical grounds. In this paper, publicly available information provided by licensed hare coursing clubs on the number of hares captured, used for coursing events, killed or injured and the percentage returned to the wild, over four coursing seasons is reviewed. A reported 19,402 hares were taken from the wild over the four coursing seasons. Whilst 19,080 hares (98%) were returned to the wild, 75 were killed by greyhounds or had to be euthanized due to their injuries. Policy makers should take animal welfare, disease control, conservation, ethics and public opinion into account and fund independent research where gaps in knowledge are identified.

**Abstract:**

Hare coursing is legal in the Republic of Ireland under licenses issued to coursing clubs but is illegal in other jurisdictions in the British Isles including Northern Ireland. Supporters of coursing maintain that coursing contributes to the conservation of the hare whilst opponents claim that coursing is cruel and the welfare of the hares is compromised. However, while the contribution of coursing to conservation has been considered, the impact of coursing on hare welfare has not been investigated. This paper reviews publicly available information from licensed hare coursing clubs over four coursing seasons, including the number of hares taken from the wild, numbers coursed, and numbers pinned to the ground by dogs, killed or injured during coursing events. In total, 19,402 hares were taken from the wild—98% of which were subsequently released back to the wild at the end of the coursing season. Almost 600 hares were pinned by greyhounds during coursing events and 75 were either killed or had to be euthanized as a result of their injuries. While the number of hares killed or injured is relatively small compared to the number caught, the welfare of all captured hares will have been compromised and has not been investigated. Policy makers must fill this knowledge gap or take a precautionary approach and further regulate or indeed prohibit the capture of hares which are otherwise fully protected.

## 1. Introduction

The Irish hare (*Lepus timidus hibernicus* Bell 1837) is widely distributed on the island of Ireland and is protected in the Republic of Ireland under the Wildlife Act 1976 and the Wildlife (Amendment) Act 2002. It is also listed on Appendix III of the Berne Convention, Annex V (a) of the EC Habitats Directive (92/43/EEC) and is recognized as an internationally important species in the Irish Red Data Book [1]. However, it is also a game or quarry species and can be hunted during an open season which runs from the 26th of September each year to the 28th of February of the following year.

Hare coursing is a widespread activity in the Republic of Ireland but it is prohibited in Northern Ireland and other jurisdictions within the British Isles (Northern Ireland: Game Preservation (Special Protection for Irish Hares) Order, Northern Ireland 2003; Scotland: Protection of Wild Mammals Act, Scotland 2002; England and Wales: Hunting Act 2004). Hare coursing is only legal in three EU Member States, Ireland, Spain and Portugal. Hare coursing involves comparing the speed and agility of greyhounds with that of the hare which is given a head start of approximately 75 m before two greyhounds are ‘slipped’ to pursue the hare. The winning dog is judged to be the one that first turns the hare, i.e., causes the hare to change direction. Each year, up to 79 coursing clubs are issued with licenses to capture hares from the wild using nets. Hares are then tagged and transported to a holding pen (or hare park) for up to two months or more during which time they will be ‘coursed’. The majority are subsequently released back to the wild at the same location as they were trapped. Up until 1993, mortality of hares in coursing was high, with thousands being killed each year [2]. In 1993, a change in legislation required that all coursing greyhounds must be muzzled. Reid et al., 2007 [2], compared the fate of hares at coursing events before and after muzzling was introduced. Unmuzzled greyhounds caused the deaths of 5171 hares over five seasons and muzzled greyhounds killed 2668 hares over 11 coursing seasons. The introduction of muzzles reduced hare mortality from 15.8% to 4.1%. However, whilst muzzling significantly reduced the numbers of hares killed or seriously injured, small numbers are still struck, pinned or tossed and either killed or injured so severely by muzzled greyhounds that they have to be euthanized by the attending veterinary practitioner.

The recent emergence of rabbit haemorrhagic disease (RHD2) in Ireland presents a significant threat to both the rabbit (*Oryctolagus cuniculus*) and hare populations and in August 2019 resulted in the suspension of licenses to take hares from the wild for hare coursing events. RHD2 was first detected in an Irish hare in August 2019 and has since been recorded in 29 rabbits in 12 counties (Carlow, Cork, Clare, Dublin, Kerry, Kildare, Leitrim, Meath, Offaly, Tipperary, Wexford and Wicklow) and five hares in two counties (Dublin and Wexford). However, the suspension of licenses was lifted in October 2019 to allow the taking of hares for coursing events in areas in which RHD2 had not yet been recorded. To reduce the risk of RHD2 spreading, the National Parks and Wildlife Service imposed 25km restriction zone around each recorded incidence of RHD2, resulting in 13 coursing clubs being unable to take hares from the wild or organise any coursing events during the 2019/2020 season.

Ongoing illegal coursing of hares also poses a significant threat to the hare population and may lead to local extinctions. Illegal hare coursing occurs all year round and targets all hares including pregnant and lactating females which results in significant animal welfare issues. Due to its very nature, illegal hare coursing is very difficult to detect, enforcement is difficult and, as a result, few convictions are achieved, although at least three convictions were recorded in 2019.

Hare coursing supporters claim that coursing makes a significant contribution to the conservation of hares in Ireland and often quote a study which used driven counts to estimate hare numbers [3]. That study estimated that overall, hares were 18 times more abundant in preserves managed by coursing clubs than in the wider countryside, and that, when habitat type was controlled for, hares were three times more abundant in coursing preserves. However, there are a number of reasons why these results may not be reliable. First, the study was carried out in just one county (Donegal in the northwest of Ireland), facilitated by the East Donegal Coursing Club. Second, the authors compared hare abundance estimates in managed preserves using 2007 data, with data for the wider countryside collected mostly in 2003, and hare population sizes are known to fluctuate widely between years. Third, the authors of the study acknowledged that although hares appeared to be maintained at high abundance in such preserves, this could simply be because coursing clubs preserved areas with high hare abundance. Therefore, the differences observed in the study may not be attributed reliably to management practices.

Although the impact of muzzling on the fate of hares has been assessed [2], the 2007 study highlighted that before the impact of hare coursing on the hare population could be accurately assessed it would be necessary to determine the overall impact of trapping and removing hares from source populations (coursing club preserves) and of returning hares to the wild after they had been coursed. This work has not been done, so it is impossible to say whether coursing contributes to the conservation of hares. In addition, although mortality of hares was reduced from 15.8% to 4.1% post-muzzling, the question remains whether it is ethically or morally acceptable that any hares are killed or severely injured for what is a minority leisure activity. Trapping, handling, transporting, keeping in captivity, coursing and finally returning them to the wild, where they were trapped, may have a negative impact on the hares on either a short-term or long-term basis. No research has been carried out to determine the impact of hare coursing on the welfare of individual hares.

The purpose of this study is to demonstrate that more information is needed to determine the impact of hare coursing on the welfare of individual hares and the Irish hare population. In the absence of such information, the precautionary principle should be applied. 

## 2. Materials and Methods

This paper reviews data provided by 72 licensed hare coursing clubs from 22 of 26 counties over four hare coursing seasons (2015/16, 2016/17, 2017/18, 2018/19) that are publicly available on forms uploaded to the website of the Irish National Parks and Wildlife Service [4]. One of the license conditions is that the coursing clubs provide certain information to the National Parks and Wildlife Service (NPWS) at the end of the coursing season. Four forms are required by the NPWS; Capture Return Form; Coursing Data Report Form; Veterinary Report Form and Hare Release Form. Not all forms were available for each club for each season (Table 1).

Using the information available on these forms, the following were calculated: total number of hares netted over the four seasons; total number of hares released over the four seasons and the numbers used at each coursing event. The numbers of hares pinned by greyhounds, injured by greyhounds, examined by a vet, euthanized as a result of their injuries, the number that died of natural causes, the number that escaped and, where possible the number unaccounted for were also calculated. The number of hares that died as a result of hare coursing was calculated as the total number of hares killed by greyhounds plus the number euthanized by the veterinary practitioner, and this was expressed as a percentage of the number of hares used.

### Post-Mortems

The results of post-mortems obtained via a freedom of information request by The Irish Council Against Blood Sports (ICABS) were examined. These related to three hares which were found by an NPWS official in a coursing compound at the end of a coursing event in Edenderry, County Offaly, in 2017 and submitted for post-mortem to ascertain cause of death.

## 3. Results

### 3.1. Hares Captured and Released

Over the four hare coursing seasons for which data were available, a minimum of 19,402 hares were netted, tagged and removed from the wild by hare coursing clubs. This ranged from 5148 in the 2015/16 season to 4593 in the 2018/19 season (average = 4851 hares per year), showing a decreasing trend in the numbers netted over the four seasons. In total, 19,080 hares (98%) were subsequently released. The minimum and maximum number of days in captivity for the four seasons ranged from 2 to 141 days, respectively, Table 2.

### 3.2. Veterinary Data

Veterinary practitioners examined 594 hares (3% of the hares that were used for coursing) that had been pinned by greyhounds, although the available forms only recorded 328 being pinned. Sixty-six of the examined hares (11%) required veterinary treatment and 27 hares (4.5%) were euthanized by the veterinary practitioner, Table 2. Forty-eight hares were killed directly by greyhounds and 22 died from unknown causes. A further 104 were recorded as having escaped or were unaccounted for, Table 2.

### 3.3. Post-Mortems

The results of the post-mortems of three hares found dead by NPWS officials at a coursing event in Edenderry, Co. Offaly, in 2017 are given in Table 3.

## 4. Discussion

Although only a small percentage (<2%) of hares taken for coursing died as a direct result of hare coursing, any leisure activity that causes unnecessary suffering or death of an animal may be considered unacceptable in a modern and progressive society. There is very little understanding of the impact of netting, handling, transport, captivity, coursing and being returned to the wild on stress or welfare among the thousands of hares that are netted each year.

Over 19,000 hares were taken from the wild and kept in captivity for up to 141 days before being released with the median maximum number of days in captivity for the 2015/16, 2016/17, 2017/18 and 2018/19 seasons being 51, 57, 57 and 28, respectively. The length of time in captivity is a concern and urgent research is needed to assess the impact of captivity on the welfare of the hares and their ability to re-assimilate when released. The licensing authority should consider introducing maximum captivity time for hares in the licensing conditions.

In August 2019, the issuing of licenses for the purpose of netting hares by coursing clubs was suspended by the Minister for Culture, Heritage and the Gaeltacht (DCHG). This was in response to the emergence of rabbit haemorrhagic disease (RHD2), in both wild rabbits and wild hares in Ireland. The hare population could be significantly impacted if the disease continues to spread. Other forms of legal hunting of hares such as Harriers and beagle packs have not been curtailed or restricted despite the risk of displacing hares and rabbits and leaving them susceptible to infection with RHD2 or encouraging spread of the disease. However, just a few weeks after licenses were suspended, the department responsible for wildlife protection lifted the suspension to allow hares to be netted for coursing in areas unaffected by the RHD2 virus. Given the lack of information about the spread of the disease, or its potential impact on the Irish hare population this decision seems premature at best and reckless at worst.

No research has been conducted to measure the impact of hare coursing on the welfare of the individual hares involved, an issue which has been largely ignored by the coursing community and the licensing authority. Netting, tagging, handling, transporting in crates, keeping in captivity with abnormal social contact, coursing itself and finally returning hares to the wild (sometimes to novel areas), may result in stress and have adverse effects on the welfare of individual hares. For example, stress may impact on a hare’s immune system and make them more susceptible to diseases such as RHD2. Long-term captivity and release to novel territories may impact on longevity and reproductive output, whilst males may have reduced ability to compete for territories and mating opportunities. When mammals are under stress, it is well known that the adrenal cortex increases production of glucocorticoids and this can be used as a non-invasive method for studying stress. This method has been tested in European brown hares, (*Lepus europeus*) [5], mountain hares (*Lepus timidus*) [6] and snowshoe hares [7]. Measuring glucocorticoids in hares netted for coursing could show the impact of coursing on the welfare of individual hares and could provide a useful tool for measuring the role of stress in hare populations in general. It is widely known that predation can have a significant impact on wild mammal populations and community processes [8]. Sherriff et al. [9] used faecal cortisol concentration to investigate the sub-lethal impacts of predation on snowshoe hares (*L. americanus*) and demonstrated that elevated, predator-induced glucocorticoid concentrations resulted in lower reproductive output. The coursing of a hare by two greyhounds mimics predation and the hare has no concept of the dogs being muzzled. It is therefore running for its life and will be subjected to the same stresses as a hare that is being chased by a predator in the wild, possibly more so with the noise of the crowd, an unfamiliar territory and very limited options for escape. In the UK, the Burns enquiry concluded that hunting with hounds (including coursing) seriously compromises the welfare of quarry species, including hares [10].

Given the paucity of information relating to the impact of hare coursing in Ireland on the welfare of individual hares and given that all of the factors listed above are known stressors to wild animals, the precautionary principle should be applied and a ban on live hare coursing introduced without delay. At the very least, independent research should be carried out to measure the impact of hare coursing on the welfare of individual hares and any impact on post-release survival or impact on future reproductive success. Setting aside animal welfare concerns, consideration should also be given to the ethical implications of live hare coursing. The Department of Agriculture, Food and the Marine (DAFM), Ireland’s competent authority for animal welfare, is in the process of introducing a new animal welfare strategy [11]. This ambitious strategy makes it clear that Ireland aspires to be recognized for its high animal welfare standards. Live hare coursing is not compatible with this policy and DAFM should recognize this. Much has been made of the contribution of live hare coursing to the conservation of the Irish hare [3] but the true contribution of coursing to conservation has been overstated and insufficient evidence is available to conclude that coursing clubs make any significant contribution. Reid et al. 2010 [3] concluded that hares were 18 times more common in preserves managed by hare coursing clubs than in the wider countryside and this is often used by the coursing community to justify their activity. However, Reid et al. (2010) [3] acknowledged confounding factors such as habitat type. When habitat was taken into account, estimated abundance was reduced from 18 times to three times that of the wider countryside. The conclusions were based on a driven count of hares which included just 135 hares. In addition, the authors compared estimations of hare abundance in preserves managed by coursing clubs using data collected in 2007 with data for the wider countryside which was mostly collected in 2003. Given the fact that wide fluctuations in Irish hare populations have been recorded e.g., [12] and in brown hares [13] it is not clear the two data sets were directly comparable. Extrapolation from this one data set should be interpreted with caution. Anecdotally, hare abundance has declined in preserves since Reid’s study and increased effort by clubs has been necessary to net sufficient hares for the coursing season with some clubs having to join together in order to catch enough hares for their coursing programme (personal communication, NPWS). If, as is often claimed, those involved in live hare coursing ‘care for the hare’, they should seek to preserve hares for their intrinsic value, postpone hare coursing until more is known about the impact of RHD2 on the hare population, and ultimately switch to lure coursing or risk coursing disappearing altogether. Further, the coursing community should support and cooperate with independent research into the impact of coursing on the welfare of hares and on local populations.

A small number of hares (N = 3) that died in captivity were submitted for post-mortem examination. Although one apparently died of natural causes, two showed signs of illness that were clearly not recognized by coursing workers which is a concern. All hares that are killed directly or indirectly as a result of coursing activities or that die whist in captivity should be submitted for post-mortem examination by the NPWS and the results made available to the public to ensure transparency and accountability, however this is not currently a license condition. The licensee and each of the clubs that are issued with licenses must arrange for a veterinary practitioner to be in attendance during all coursing and trial meetings. The veterinary practitioner is responsible for completing the Veterinary Report Form in which they should include a report on the general health of the hares and on any injuries or deaths of hares that occur during or following completion of the coursing meeting and before they are returned to the wild. Hares that become sick or injured in captivity are not permitted to be used for coursing, but the license does not prescribe what should be done with sick or injured hares. Licenses should make it clear that clubs should seek veterinary treatment for any sick or injured hare.

Illegal (i.e., unlicensed) hare coursing is still widespread in Ireland, occurs at all times of the year (not just during the Open Season which runs from October to February) and the lurchers or greyhounds used are not muzzled. Hares are regularly killed, including pregnant and lactating females. This practice is likely to have a negative impact not only on the welfare of the hares but also on local populations. Combined with the potential impact of legal coursing on individual hares and local hare populations, hare coursing both legal and illegal may have a negative impact on hares. The precautionary principle should be applied and licenses suspended until independent research has been carried out on the impact of hare coursing on the welfare of hares and hare populations. There is no evidence that, as is often claimed by the Irish Coursing Club, prohibition of licensed hare coursing would result in more illegal hare coursing, but more resources and collaborative working between relevant stake holders including the NPWS, An Garda Sióchana, DAFM and animal welfare groups should be utilized to target illegal hare coursing and other wildlife crime.

## Figures and Tables

**Table 1 animals-10-00163-t001:** Details of the forms available on the website of the National Parks and Wildlife Service. All of the information listed in the table was publicly available with the exception of the location of netting and release of hares which were redacted by National Parks and Wildlife Service (NPWS). Any information which identified individuals (e.g., name of veterinary practitioner) was also redacted.

Name of Form	Information on Form
Capture Return Form	Season Name of club Date hares netted Location of netting Number of hares netted
Coursing Data Report	Season Name of club Event date No. of competitive courses run No. of reserve courses run No. of hares pinned * No. of hares examined by vet No. of hares requiring treatment from vet No. of hares euthanized by vet No. of hares that died from injuries No. of hares that died of natural causes Date hares released No. of hares released
Veterinary Report	Season Name of club Date (s) of events No. of hares used each day No of hares examined for injuries No. of hares confirmed injured No. of hares euthanized No. of hares that died of injuries No. of hares that died of natural causes Details and results of any post-mortems
Hare Release Form	Season Name of club Date of release Location of release No. of hares released

* Pinned = held down by muzzled greyhound.

**Table 2 animals-10-00163-t002:** Details of the numbers of hares netted and taken from the wild for hare coursing in Ireland over four hare coursing seasons, the number of coursing events, time in captivity and fate of hares used.

Season	No. of Coursing Events	No. of Hares Taken from Wild	Min Length of Time in Captivity (Days)	Max Length Time in Captivity (Days)	No. of Hares Used	No. of Hares Pinned	No. of Pinned Hares Examined by Veterinary Practitioner	No. of Pinned Hares Requiring Treatment from Veterinary Practitioner	No. of Hares Killed by Greyhound	No. of Injured Hares Euthanized by Veterinary Practitioner	No. of Hares Released	No. of Hares Escaped or not Accounted for	No. of Hares Died (Cause Unknown)
**2018/2019**	185	4593	6	107	4592	130	127	19	5	8	4579	5	5
**2017/2018**	166	4756	4	141	4738	176	168	20	10	3	4626	55	8
**2016/2017**	148	4905	2	126	4882	17	132	0	14	3	4794	22	5
**2015/2016**	136	5148	4	94	5146	5	167	27	19	13	5081	22	4
**Total**		19,402			19,358	328	594	66	48	27	19,080	104	22

The number of pinned hares examined by veterinary practitioner differs from the numbers recorded as pinned in both 2015/16 and 2016/17, which suggests that the numbers pinned were not accurately recorded for these two seasons.

**Table 3 animals-10-00163-t003:** Results of post-mortems of three hares found dead by NPWS in hare compound at a coursing club in Edenderry, Co. Offaly, in 2017.

Hare	Post-Mortem Results
Hare 1	Underweight, some dermatitis around mouth/jaw. No other visible wounds or external injuries, no evidence of internal bruising or injury on internal examination. Carcass mild-moderate dehydration, stomach and bowel less than half full—large grass and oat particles, poorly chewed, found in stomach. No visible lesions indicating infectious disease in abdomen or thorax. Oral examination revealed one molar tooth absent with ulceration of underlying gum and abscess in jaw. Diagnosis: dental disease with chronic infection of jaw.
Hare 2	Average body condition with no external wounds or bruising under the skin, noticeably grey around muzzle—older hare? No visible lesions or abscesses detected in thorax or abdomen but kidneys small and firm (both < 2 cm) but appear normal on cross-section. Heart and lungs grossly normal, no evidence of pneumonia, lung worm or cardiomyopathy. Stomach full, contents appear normal consistency with evidence of grass and oats ingested within previous 24 h, no evidence of toxic plants ingested. Small bowel, within normal limits, soft caetotrophs in large bowel. Diagnosis: presumed natural causes.
Hare 3	Below average body condition, no external wounds identified and no bruising under skin. Grass particles only in stomach, no abnormal lesions in small bowel, normal faecal pellets in large bowel, 4 small (2–3 mm) nodules with hardened pus and calcification in liver indicative chronic abscessation/scarring. Evidence of pleurisy with some scarring/adhesions between lung, diaphragm and pleura, old calcified abscesses in lungs similar to liver lesions, lungs fibrous on cross-section consistent with long standing infection and scarring. Diagnosis: chronic pleuropneumonia and hepatitis.

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
