# Peer review of "Welfare Implications for Hares, Lepus timidus hibernicus, Taken from the Wild for Licensed Hare Coursing in Ireland"

_animals, 2020, doi:10.3390/ani10010163_

Round 1
Reviewer 1 Report
13 “released from captivity” rather than “returned to the wild”?
14-15 Arguably Reid et al 2007? Suggest “little research”
15 Capture? Keeping? Chase?
17 Are there data on non-vet euthanasia?
19 Remove “animal welfare” as it is at the start of the sentence.
21 See 13 above
22-23 Give dates
24 Euthanized or euthanased?
32 See 14-15 above
32-33 Give dates
36 Were reported as?
38-39 This is unclear: the welfare of the killed hares? Or all hares caught?
47 Wildlife (Amendment) Act??
50 Where is it classified as a game or quarry species?
53 Define or remove “common”
57 How many EU states have hares?
57-59 Are the dogs being compared with each other and/or with the hare?
61-63 In terms of welfare, this must be huge?
69 Pinned, not pined
74 Reference needed
79-80 Does “legal” capture pose any threat?
83 Reference needed
89 Insert “after” or remove comma
92 Reference needed
93 Typo: “set”
96 Remove “typical”
97 Data are plural: data “were collected”
105 Or otherwise
112 The description of the purpose of the study is too conclusive and presumptive. Is it possible to undertake a study like this to demonstrate that more information is needed? Surely the study revealed that more information is needed.
119 “(reference)” – correct
121-122 “Hare Release Form” is missing - ?
128 Consider giving a definition of Pinned earlier in the paper
129 Who is “we”? Most of the paper is in the passive voice
129-131 Use number not no.
137 Are the results of three post-mortems from one event in any way significant?
141 See 138 – post-mortem or post mortem?
147-149 The median / mean duration would be informative
151-153 Number of hares pinned is less than number of pinned hares examined by a vet – this is explained in 155-157, but is confusing in the table.
157 Make it clear that the percentage is of 594
171-175 Suggest reworking this paragraph
179 Typo – extra space
181-182 Harriers (capital) and beagle (non-capital) – amend
185-187 Too emotive – amend
190 Grammar is incorrect and this should, if possible, be supported with evidence
203-206 Long and confused sentence – amend
214-215 I feel this sentiment could be worded less stridently and still make the same point.
217-218 The author should explain the separation, or otherwise, between animal welfare and ethical concerns
220 Missing reference
221-222 This is too strongly worded and should be amended
250 Unless there is evidence presented, “regularly” should be amended
Reviewer 2 Report
I think that this manuscript has merit, but the author falls into the trap of letting his personal views colour the Discussion and fails to set out clearly what options the authority responsible for regulating coursing could do to move either rapidly, or more slowly but by logical steps to reducing welfare issues associated with this sport. The assumption that this type of recreational pursuit will have to end is a rather naive position to arrive at. The author has pointed out that by virtue of the quality of licence returns, and already acknowledged illegal activities, that making coursing illegal will improve welfare outcomes for wild hares. In all likelihood this would have the opposite effect, as illegal activities with no regulation or oversight by vets would become the new norm.
Several sentences and paragraphs require references to substantiate them (e.g. L73-77) and those sentences where the author has forgotten to add intended references (L119, L220).
The paragraph from L84-98 is better suited in the Discussion, and a very similar text already exists there. Choose one or the other, but delete the paragraph from the Introduction.
There is another question that needs to be addressed in this whole process. That is if hares are caught but not used, and then returned to the wild do any of them get recaptured and suffer a second (or multiple) round of captivity. It is not clear if the marking process required by regulator is a permanent one (e.g. microchip) that could then be used to exclude those animals that have already been 'used' once, or if the marking is temporary and only suitable for the duration of captivity.
All mean values given should include SD or SE values.
Four application forms are referred to in the para L117-123, but only three are listed. The 4th - Hare release form is missing.
Table 2 would be improved by including percentages after the nominal values, but care will be need to indicate that some percentages will be of sub-tallies and not the grand total. The order of the results in the table should be reversed so that the data are presented from oldest to most recent.
If the author wants to make some useful suggestions in the Discussion it would be worthwhile clearly setting out the types of research/experiments that would be required to provide empirical evidence that will conclusively show that coursing has a negligible and acceptable welfare risk, or that things need to change (up to and including a full cessation of coursing).
I think there is also a case for looking in more detail at the trend for hares to be captured in one county and shipped to another due to current real or perceived shortfalls in supply (i.e. wild population). This would suggest that the current level of harvest may not be sustainable in some counties, and also that returning hares not used or which survive a coursing event, becomes much more difficult and or costly.
The one piece of evidence that I found really staggering was that hares were being kept up to 171 days in captivity. Even if animals that had been kept this long were used in coursing it is difficult to argue that this was any real test of greyhound ability, and it also means that any return of such hares to the wild might not result in assimilation back to the wild. Arguing for shorter maximum periods of captivity, regular vet checks of animals while being held, and urgent research to confirm the survival and re-integration of hares back into the wild might just provide some case for the continuation of coursing into the future. The sport/industry could do little else but agree to such changes and research, and if they can't contribute to the cost of those changes then that alone might be the key element necessary to see, what for some sections of the community is the only acceptable outcome.
Round 2
Reviewer 2 Report
A much improved manuscript now that the personal emotion has been removed for the most part. A few minor edits:
L28 and L53 change 'despite being' to 'and is'
L56 delete 'Indeed' and strt the sentence with the word 'Hare'.
L58 Delete the word 'chase'
L64 Add the word 'in' after 'change'
L69 'pinned' not 'pined'
L102 Add 'determine' after 'assessed it would be necessary'
L119 Reference still required.
L147 change to read 'netted, and ...'
L222 Reference still required.
Author Response
Reply to the Review Report (Reviewer 2).
I am grateful to Reviewer 2 for the comments on the latest version of the MS.
Comments and suggestions
L28 and L53 – I have changed ‘despite being’ to ‘and is’
L56 – I have deleted ‘Indeed’ and have started the sentence with ‘Hare….’
L58 – I have deleted ‘chase’ and retained ‘pursue’
L64 – I have added ‘in’ after ‘change’
L69 – I have corrected spelling of ‘pinned’
L102 – I have added ‘determine’ after ‘assessed it would be necessary’
L119 – I have added the reference - 4. National Parks and Wildlife Service, Hare Coursing https://www.npws.ie/licences/hare-coursing
L147 – I could not find where in L147 the reviewer intended the change to be made. Perhaps it will be picked up in editing.
L222 – the government strategy will be launched in January 2020 following a consultation which was published in October 2018. If the MS is published after the launch I can provide a reference (link to DAFM website) for the strategy – in the meantime I have given a link to the publication of the draft strategy and have changed the text to say that the government is in the process of introducing a new animal welfare strategy.
